# Multi-Objective Non-parametric Sequential Prediction

**Guy Uziel**
Computer Science Department
Technion - Israel Institute of Technology
guziel@cs.technion.ac.il

**Ran El-Yaniv**
Computer Science Department
Technion - Israel Institute of Technology
rani@cs.technion.ac.il

## Abstract

Online-learning research has mainly been focusing on minimizing one objective function. In many real-world applications, however, several objective functions have to be considered simultaneously. Recently, an algorithm for dealing with several objective functions in the i.i.d. case has been presented. In this paper, we extend the multi-objective framework to the case of stationary and ergodic processes, thus allowing dependencies among observations. We first identify an asymptomatic lower bound for any prediction strategy and then present an algorithm whose predictions achieve the optimal solution while fulfilling any continuous and convex constraining criterion.

## 1 Introduction

In the traditional online learning setting, and in particular in sequential prediction under uncertainty, the learner is evaluated by a single loss function that is not completely known at each iteration [6]. When dealing with multiple objectives, since it is impossible to simultaneously minimize all of the objectives, one objective is chosen as the main function to minimize, leaving the others to be bound by pre-defined thresholds. Methods for dealing with one objective function can be transformed to deal with several objective functions by giving each objective a pre-defined weight. The difficulty, however, lies in assigning an appropriate weight to each objective in order to keep the objectives below a given threshold. This approach is very problematic in real world applications, where the player is required to to satisfy certain constraints. For example, in online portfolio selection [4], the player may want to maximize wealth while keeping the risk (i.e., variance) contained below a certain threshold. Another example is the Neyman-Pearson (NP) classification paradigm (see, e.g., [19]) (which extends the objective in classical binary classification) where the goal is to learn a classifier achieving low classification error whose type I error is kept below a given threshold.

In the adversarial setting it is known that multiple-objective is generally impossible when the constraints are unknown a-priory [18]. In the stochastic setting, Mahdavi et al. [17] proposed a framework for dealing with multiple objectives in the i.i.d. case. They proved that if there exists a solution that minimizes the main objective function while keeping the other objectives below given thresholds, then their algorithm will converge to the optimal solution.

In this work, we study online prediction with multiple objectives but now consider the challenging general case where the unknown underlying process is stationary and ergodic, thus allowing observations to depend on each other arbitrarily. The (single-objective) sequential prediction under stationary and ergodic sources, has been considered in many papers and in various application domains. For example, in online portfolio selection, [12, 9, 10] proposed non-parametric online strategies that guarantee, under mild conditions, the best possible outcome. Another interesting example in this regard is the work on time-series prediction by [2, 8, 3]. A common theme to all these results is that the asymptotically optimal strategies are constructed by combining the predictions of many simple

experts. The above strategies use a countably infinite set of experts, and the guarantees provided for these strategies are always asymptotic. This is no coincidence, as it is well known that finite sample guarantees for these methods cannot be achieved without additional strong assumptions on the source distribution [7, 16]. Approximate implementations of non-parametric strategies (which apply only a finite set of experts), however, turn out to work exceptionally well and, despite the inevitable approximation, are reported [11, 10, 9] to significantly outperform strategies designed to work in an adversarial, no-regret setting, in various domains.

The algorithm presented in this paper utilizes as a sub-routine the Weak Aggregating Algorithm (WAA) of [21], and [13] to handle multiple objectives. While we discuss here the case of only two objective functions, our theorems can be extended easily to any fixed number of functions.

## 2   Problem Formulation

We consider the following prediction game. Let $\mathcal{X} \triangleq [-D, D]^d \subset \mathbb{R}^d$ be a compact *observation space* where $D > 0$. At each round, $n = 1, 2, \ldots$, the player is required to make a prediction $y_n \in \mathcal{Y}$, where $\mathcal{Y} \subset \mathbb{R}^m$ is a compact and convex set, based on past observations, $X_1^{n-1} \triangleq (x_1, \ldots, x_{n-1})$ and, $x_i \in \mathcal{X}$ ($X_1^0$ is the empty observation). After making the prediction $y_n$, the observation $x_n$ is revealed and the player suffers two losses, $u(y_n, x_n)$ and $c(y_n, x_n)$, where $u$ and $c$ are real-valued continuous functions and convex w.r.t. their first argument. We view the player's prediction strategy as a sequence $\mathcal{S} \triangleq \{S_n\}_{n=1}^{\infty}$ of forecasting functions $S_n : \mathcal{X}^{(n-1)} \to \mathcal{Y}$; that is, the player's prediction at round $n$ is given by $S_n(X_1^{n-1})$ (for brevity, we denote $S(X_1^{n-1})$). Throughout the paper we assume that $x_1, x_2, \ldots$ are realizations of random variables $X_1, X_2, \ldots$ such that the stochastic process $(X_n)_{-\infty}^{\infty}$ is jointly stationary and ergodic and $\mathbb{P}(X_i \in \mathcal{X}) = 1$. The player's goal is to play the game with a strategy that minimizes the average $u$-loss, $\frac{1}{N} \sum_{i=1}^{N} u(S(X_1^{i-1}), x_i)$, while keeping the average $c$-loss $\frac{1}{N} \sum_{i=1}^{N} c(S(X_1^{i-1}), x_i)$ bounded below a prescribed threshold $\gamma$. Formally, we define the following:

**Definition 1** ($\gamma$-bounded strategy)**.** *A prediction strategy $\mathcal{S}$ will be called $\gamma$-bounded if*

$$\limsup_{N \to \infty} \left( \frac{1}{N} \sum_{i=1}^{N} c(S(X_1^{i-1}), X_i) \right) \leq \gamma$$

*almost surely. The set of all $\gamma$-bounded strategies will be denoted $\mathcal{S}_\gamma$.*

The well known result of [1] states that for the single objective case the best possible outcome is $\mathbb{E} \left[ \max_{y \in \mathcal{Y}()} \mathbb{E}_{\mathbb{P}_\infty} [u(y, X_0)] \right]$ where $\mathbb{P}_\infty$ is the regular conditional probability distribution of $X_0$ given $\mathcal{F}_\infty$ (the $\sigma$-algebra generated by the infinite past $X_{-1}, X_{-2}, \ldots$) and the maximization is over the $\mathcal{F}_\infty$-measurable functions. This motivates us to define the following:

**Definition 2** ($\gamma$-feasible process)**.** *We say that the stationary and ergodic process $\{X_i\}_{-\infty}^{\infty}$ is $\gamma$-feasible w.r.t. the functions $u$ and $c$, if for a threshold $\gamma > 0$, there exists some $y' \in \mathcal{Y}()$ such that $\mathbb{E}[c(y', X_0)] < \gamma$.*

If $\gamma$-feasibility holds, then we will denote by $y_\infty^*$ ($y_\infty^*$ is not necessarily unique) the minimizer of the following minimization problem:

$$\begin{aligned} \underset{y \in \mathcal{Y}()}{\text{minimize}} \quad & \mathbb{E}\left[u(y, X_0)\right] \\ \text{subject to} \quad & \mathbb{E}\left[c(y, X_0)\right] \leq \gamma, \end{aligned} \tag{1}$$

(1) and we define the $\gamma$-*feasible optimal value* as

$$\mathcal{V}^* = \mathbb{E}\left[\mathbb{E}_{\mathbb{P}_\infty}\left[u(y_\infty^*, X_0)\right]\right].$$

Note that problem (1) is a convex minimization problem over $\mathcal{Y}()$. Therefore, the problem is equivalent to finding the saddle point of the Lagrangian function [15], namely,

$$\min_{y \in \mathcal{Y}()} \max_{\lambda \in \mathbb{R}^+} \mathcal{L}(y, \lambda),$$

where the Lagrangian is

$$\mathcal{L}(y, \lambda) \triangleq \left(\mathbb{E}\left[u(y, X_0)\right] + \lambda \left(\mathbb{E}\left[c(y, X_0)\right] - \gamma\right)\right).$$

We denote the optimal dual by $\lambda_\infty^*$ and assume that $\mathcal{L}$ can be maximize by a unique $\mathcal{F}_\infty$-measurable function, $\lambda_\infty^*()$. Moreover, we set a constant $\lambda_{\max}$ such that $\lambda_{\max} > \lambda_\infty^*()$ $\mathbb{P}_\infty$-a.s., and set $\Lambda \triangleq [0, \lambda_{\max}]$. We also define the *instantaneous Lagrangian function* as

$$l(y, \lambda, x) \triangleq u(y, x) + \lambda \left( c(y, x) - \gamma \right). \tag{2}$$

In Brief, we are seeking a strategy $\mathcal{S} \in \mathcal{S}_\gamma$ that is as good as any other $\gamma$-bounded strategy, in terms of the average $u$-loss, when the underlying process is $\gamma$-feasible. Such a strategy will be called $\gamma$-universal.

# 3   Optimality of $\mathcal{V}^*$

In this section, we show that the average $u$-loss of any $\gamma$-bounded prediction strategy cannot be smaller than $\mathcal{V}^*$, the $\gamma$-feasible optimal value. This result is a generalization of the well-known result of [1] regarding the best possible outcome under a single objective. Before stating and proving this optimality result, we state three lemmas that will be used repeatedly in this paper. The first lemma is known as Breiman's generalized ergodic theorem. The second and the third lemmas concern the continuity of the saddle point w.r.t. the probability distribution, their proofs appear in the supplementary material.

**Lemma 1** (Ergodicity, [5]). *Let $\mathbf{X} = \{X_i\}_{-\infty}^\infty$ be a stationary and ergodic process. For each positive integer $i$, let $T_i$ denote the operator that shifts any sequence by $i$ places to the left. Let $f_1, f_2, \ldots$ be a sequence of real-valued functions such that $\lim_{n\to\infty} f_n(\mathbf{X}) = f(\mathbf{X})$ almost surely, for some function $f$. Assume that $\mathbb{E} \sup_n |f_n(\mathbf{X})| < \infty$. Then,*

$$\lim_{n\to\infty} \frac{1}{n} \sum_{i=1}^n f_i(T^i \mathbf{X}) = \mathbb{E} f(\mathbf{X})$$

*almost surely.*

**Lemma 2** (Continuity and Minimax). *Let $\mathcal{Y}, \Lambda, \mathcal{X}$ be compact real spaces. $l : \mathcal{Y} \times \Lambda \times \mathcal{X} \to \mathbb{R}$ be a continuous function. Denote by $\mathbb{P}(\mathcal{X})$ the space of all probability measures on $\mathcal{X}$ (equipped with the topology of weak-convergence). Then the following function $L^* : \mathbb{P}(\mathcal{X}) \to \mathbb{R}$ is continuous*

$$L^*(\mathbb{Q}) = \inf_{y \in \mathcal{Y}} \sup_{\lambda \in \Lambda} \mathbb{E}_\mathbb{Q} \left[ l(y, \lambda, x) \right]. \tag{3}$$

*Moreover, for any $\mathbb{Q} \in \mathbb{P}(\mathcal{X})$,*

$$\inf_{y \in \mathcal{Y}} \sup_{\lambda \in \Lambda} \mathbb{E}_\mathbb{Q} \left[ l(y, \lambda, x) \right] = \sup_{\lambda \in \Lambda} \inf_{y \in \mathcal{Y}} \mathbb{E}_\mathbb{Q} \left[ l(y, \lambda, x) \right].$$

**Lemma 3** (Continuity of the optimal selection). *Let $\mathcal{Y}, \Lambda, \mathcal{X}$ be compact real spaces. Then, there exist two measurable selection functions $h^X, h^\lambda$ such that*

$$h^y(\mathbb{Q}) \in \arg\min_{y \in \mathcal{Y}} \left( \max_{\lambda \in \Lambda} \mathbb{E}_\mathbb{Q} \left[ l(y, \lambda, x) \right] \right), \quad h^\lambda(\mathbb{Q}) \in \arg\max_{\lambda \in \Lambda} \left( \min_{y \in \mathcal{Y}} \mathbb{E}_\mathbb{Q} \left[ l(y, \lambda, x) \right] \right)$$

*for any $\mathbb{Q} \in \mathbb{P}(\mathcal{X})$. Moreover, let $L^*$ be as defined in Equation (3). Then, the set*

$$Gr(L^*) \triangleq \{ (u^*, v^*, \mathbb{Q}) \mid u^* \in h^y(\mathbb{Q}), v^* \in h^\lambda(\mathbb{Q}), \mathbb{Q} \in \mathbb{P}(\mathcal{X}) \},$$

*is closed in $\mathcal{Y} \times \Lambda \times \mathbb{P}(\mathcal{X})$.*

The importance of Lemma 3 stems from the fact that it proves the continuity properties of the multi-valued correspondences $\mathbb{Q} \to h^y(\mathbb{Q})$ and $\mathbb{Q} \to h^\lambda(\mathbb{Q})$. This leads to the knowledge that if for the limiting distribution, $\mathbb{Q}_\infty$, the optimal set is a singleton, then $\mathbb{Q} \to h^y(\mathbb{Q})$ and $\mathbb{Q} \to h^\lambda(\mathbb{Q})$ are continuous in $\mathbb{Q}_\infty$. We are now ready to prove the optimality of $\mathcal{V}^*$.

**Theorem 1** (Optimality of $\mathcal{V}^*$). *Let $\{X_i\}_{-\infty}^\infty$ be a $\gamma$-feasible process. Then, for any strategy $\mathcal{S} \in \mathcal{S}_\gamma$, the following holds a.s.*

$$\liminf_{N\to\infty} \frac{1}{N} \sum_{i=1}^N u(S(X_1^{i-1}), X_i) \geq \mathcal{V}^*.$$

*Proof.* For any given strategy $\mathcal{S} \in \mathcal{S}_\gamma$, we will look at the following sequence:

$$\frac{1}{N} \sum_{i=1}^{N} l(S(X_1^{i-1}), \tilde{\lambda}_i^*, X_i). \tag{4}$$

where $\tilde{\lambda}_i^* \in h^\lambda(\mathbb{P}_{X_i | X_1^{i-1}})$ Observe that

$$(4) = \frac{1}{N} \sum_{i=1}^{N} \left( l(S(X_1^{i-1}), \tilde{\lambda}_i^*, X_i) - \mathbb{E}\left[ l(S(X_1^{i-1}), \tilde{\lambda}_i^*, X_i) \mid X_1^{i-1} \right] \right)$$

$$+ \frac{1}{N} \sum_{i=1}^{N} \mathbb{E}\left[ l(S(X_1^{i-1}), \tilde{\lambda}_i^*, X_i) \mid X_1^{i-1} \right].$$

Since $A_i = l(S(X_1^{i-1}), \tilde{\lambda}_i^*, X_i) - \mathbb{E}\left[ l(S(X_1^{i-1}), \tilde{\lambda}_i^*, X_i) \mid X_1^{i-1} \right]$ is a martingale difference sequence, the last summand converges to 0 a.s., by the strong law of large numbers (see, e.g., [20]). Therefore,

$$\liminf_{N \to \infty} \frac{1}{N} \sum_{i=1}^{N} l(S(X_1^{i-1}), \tilde{\lambda}_i^*, X_i) = \liminf_{N \to \infty} \frac{1}{N} \sum_{i=1}^{N} \mathbb{E}\left[ l(S(X_1^{i-1}), \tilde{\lambda}_i^*, X_i) \mid X_1^{i-1} \right]$$

$$\geq \liminf_{N \to \infty} \frac{1}{N} \sum_{i=1}^{N} \min_{y \in \mathcal{Y}()} \mathbb{E}\left[ l(y, \tilde{\lambda}_i^*, X_i) \mid X_1^{i-1} \right], \tag{5}$$

where the minimum is taken w.r.t. all the $\sigma(X_1^{i-1})$-measurable functions. Because the process is stationary, we get for $\hat{\lambda}_i^* \in h^\lambda(\mathbb{P}_{X_0 | X_{1-i}^{-1}})$,

$$(5) = \liminf_{N \to \infty} \frac{1}{N} \sum_{i=1}^{N} \min_{y \in \mathcal{Y}()} \mathbb{E}\left[ l(y, \hat{\lambda}_i^*, X_0) \mid X_{1-i}^{-1} \right] = \liminf_{N \to \infty} \frac{1}{N} \sum_{i=1}^{N} L^*(\mathbb{P}_{X_0 | X_{1-i}^{-1}}). \tag{6}$$

Using Levy's zero-one law, $\mathbb{P}_{X_0 | X_{1-i}^{-1}} \to \mathbb{P}_\infty$ weakly as $i$ approaches $\infty$ and from Lemma 2 we know that $L^*$ is continuous. Therefore, we can apply Lemma 1 and get that a.s.

$$(6) = \mathbb{E}[L^*(\mathbb{P}_\infty)] = \mathbb{E}\left[\mathbb{E}_{\mathbb{P}_\infty}[l(y_\infty^*, \lambda_\infty^*, X_0)]\right] = \mathbb{E}[\mathcal{L}(y_\infty^*, \lambda_\infty^*, X_0)]. \tag{7}$$

Note also, that due to the complementary slackness condition of the optimal solution, i.e., $\mathbb{E}[\lambda_\infty^*(\mathbb{E}_{\mathbb{P}_\infty}[c(y_\infty^*, X_0)] - \gamma)] = 0$, we get

$$(7) = \mathbb{E}\left[\mathbb{E}_{\mathbb{P}_\infty}[u(y_\infty^*, X_0)]\right] = \mathcal{V}^*.$$

From the uniqueness of $\lambda_\infty^*$, and using Lemma 3 $\hat{\lambda}_i^* \to \lambda_\infty^*$ as $i$ approaches $\infty$. Moreover, since $l$ is continuous on a compact set, $l$ is also uniformly continuous. Therefore, for any given $\epsilon > 0$, there exists $\delta > 0$, such that if $|\lambda' - \lambda| < \delta$, then

$$|l(y, \lambda', x) - l(y, \lambda, x)| < \epsilon$$

for any $y \in \mathcal{Y}$ and $x \in \mathcal{X}$. Therefore, there exists $i_0$ such that if $i > i_0$ then $|l(y, \hat{\lambda}_i^*, x) - l(y, \lambda_\infty^*, x)| < \epsilon$ for any $y \in \mathcal{Y}$ and $x \in \mathcal{X}$. Thus,

$$\liminf_{N \to \infty} \frac{1}{N} \sum_{i=1}^{N} l(S(X_1^{i-1}), \lambda_\infty^*, X_i) - \liminf_{N \to \infty} \frac{1}{N} \sum_{i=1}^{N} l(S(X_1^{i-1}), \hat{\lambda}_i^*, X_i)$$

$$= \liminf_{N \to \infty} \frac{1}{N} \sum_{i=1}^{N} l(S(X_1^{i-1}), \lambda_\infty^*, X_i) + \limsup_{N \to \infty} \frac{1}{N} \sum_{i=1}^{N} -l(S(X_1^{i-1}), \hat{\lambda}_i^*, X_i)$$

$$\geq \liminf_{N \to \infty} \frac{1}{N} \sum_{i=1}^{N} l(S(X_1^{i-1}), \hat{\lambda}_i^*, X_i) - \frac{1}{N} \sum_{i=1}^{N} l(S(X_1^{i-1}), \lambda_\infty^*, X_i) \geq -\epsilon \ a.s.,$$

---

**Algorithm 1** Minimax Histogram Based Aggregation (MHA)

---

**Input:** Countable set of experts $\{H_{k,h}\}$, $y_0 \in \mathcal{Y}$, $\lambda_0 \in \Lambda$, initial probability $\{\alpha_{k,h}\}$,
**For** $n = 0$ **to** $\infty$
   Play $y_n, \lambda_n$.
   Nature reveals $x_n$
   Suffer loss $l(y_n, \lambda_n, x_n)$.
   Update the cumulative loss of the experts

$$l_{y,n}^{k,h} \triangleq \sum_{i=0}^{n} l(y_{k,h}^i, \lambda_i, x_i) \qquad l_{\lambda,n}^{k,h} \triangleq \sum_{i=0}^{n} l(y_i, \lambda_{k,h}^i, x_i)$$

   Update experts' weights

$$w_n^{y,(k,h)} \triangleq \alpha_{k,h} \exp\left(-\frac{1}{\sqrt{n}} l_{y,n}^{k,h}\right) \qquad p_{n+1}^{y,(k,h)} \triangleq \frac{w_{n+1}^{y,(k,h)}}{\sum_{h=1}^{\infty}\sum_{k=1}^{\infty} w_{n+1}^{y,(k,h)}}$$

   Update experts' weights $w_{n+1}^{\lambda,(k,h)}$

$$w_{n+1}^{\lambda,(k,h)} \triangleq \alpha_{k,h} \exp\left(\frac{1}{\sqrt{n}} l_{\lambda,n}^{k,h}\right) \qquad p_{n+1}^{\lambda,(k,h)} = \frac{w_{n+1}^{\lambda,(k,h)}}{\sum_{h=1}^{\infty}\sum_{k=1}^{\infty} w_{n+1}^{\lambda,(k,h)}}$$

   Choose $y_{n+1}$ and $\lambda_{n+1}$ as follows

$$y_{n+1} = \sum_{k,h} p_{n+1}^{y,(k,h)} y_{k,h}^{n+1} \qquad \lambda_{n+1} = \sum_{k,h} p_{n+1}^{\lambda,(k,h)} \lambda_{k,h}^{n+1}$$

**End For**

---

and since $\epsilon$ is arbitrary,

$$\liminf_{N\to\infty} \frac{1}{N}\sum_{i=1}^{N} l(S(X_1^{i-1}), \lambda_\infty^*, X_i) \geq \liminf_{N\to\infty} \frac{1}{N}\sum_{i=1}^{N} l(S(X_1^{i-1}), \hat{\lambda}_i^*, X_i).$$

Therefore we can conclude that

$$\liminf_{N\to\infty} \frac{1}{N}\sum_{i=1}^{N} l(S(X_1^{i-1}), \lambda_\infty^*, X_i) \geq \mathcal{V}^* \ a.s.$$

We finish the proof by noticing that since $\mathcal{S} \in \mathcal{S}_\gamma$, then by definition

$$\limsup_{N\to\infty} \frac{1}{N}\sum_{i=1}^{N} c(S(X_1^{i-1}), X_i) \leq \gamma \ a.s.$$

and since $\lambda_\infty^*$ is non negative, we will get the desired result. $\square$

The above lemma also provides the motivation to find the saddle point of the Lagrangian $\mathcal{L}$. Therefore, for the reminder of the paper we will use the loss function $l$ as defined in Equation 2.

## 4 Minimax Histogram Based Aggregation

We are now ready to present our algorithm *Minimax Histogram based Aggregation (MHA)* and prove that its predictions are as good as the best strategy.

By Theorem 1 we can restate our goal: find a prediction strategy $\mathcal{S} \in \mathcal{S}_\gamma$ such that for any $\gamma$-feasible process $\{X_i\}_{-\infty}^{\infty}$ the following holds:

$$\lim_{N\to\infty} \frac{1}{N}\sum_{i=1}^{N} u(S(X_1^{i-1}), X_i) = \mathcal{V}^* \ a.s.$$

Such a strategy will be called $\gamma$-*universal*. We do so by maintaining a countable set of experts $\{H_{k,h}\}$ $k, h = 1, 2, \ldots$, which are constructed in a similar manner to the experts used in [10]. Each expert is defined using a histogram which gets finer as $h$ grows, allowing us to construct an empirical measure on $\mathcal{X}$. An expert $H_{k,h}$ therefore outputs a pair $(y_{k,h}^i, \lambda_{k,h}^i) \in \mathcal{Y} \times \Lambda$ at round $i$. This pair is the minimax w.r.t. its empirical measure. We show that those empirical measures converge weakly to $\mathbb{P}_\infty$, thus, the experts' prediction will converge to $\mathcal{V}^*$. Our algorithm outputs at round $i$ a pair $(y_i, \lambda_i) \in \mathcal{Y} \times \Lambda$ where the sequence of predictions $y_1, y_2, \ldots$ tries to minimize the average loss $\frac{1}{N} \sum_{i=1}^N l(y, \lambda_i, x_i)$ and the sequence of predictions $\lambda_1, \lambda_2, \ldots$ tries to maximize the average loss $\frac{1}{N} \sum_{i=1}^N l(y_i, \lambda, x_i)$. Each of $y_i$ and $\lambda_i$ is the aggregation of predictions $y_{k,h}^i$ and $\lambda_{k,h}^i$, $k, h = 1, 2, \ldots$, respectively. In order to ensure that the performance of MHA will be as good as any other expert for both the $y$ and the $\lambda$ predictions, we apply the Weak Aggregating Algorithm of [21], and [13] twice alternately. Theorem 2 states that the selection of points made by the experts above converges to the optimal solution, the proof of Theorem 2 and the explicit construction of the experts appears in the supplementary material. Then, in Theorem 3 we prove that MHA applied on the experts defined in Theorem 2 generates a sequence of predictions that is $\gamma$-bounded and as good as any other strategy w.r.t. any $\gamma$-feasible process.

**Theorem 2.** *Assume that $\{X_i\}_{-\infty}^\infty$ is a $\gamma$-feasible process. Then, it is possible to construct a countable set of experts $\{H_{k,h}\}$ for which*

$$\lim_{k \to \infty} \lim_{h \to \infty} \lim_{n \to \infty} \frac{1}{N} \sum_{i=1}^N l(y_{k,h}^i, \lambda_{k,h}^i, X_i) = \mathcal{V}^* \quad a.s.,$$

*where $(y_{k,h}^i, \lambda_{k,h}^i)$ are the predictions made by expert $H_{k,h}$ at round $i$.*

Before stating the main theorem regarding MHA, we state the following lemma (the proof appears in the supplementary material), which is used in the proof of the main result regarding MHA.

**Lemma 4.** *Let $\{H_{k,h}\}$ be a countable set of experts as defined in the proof of Theorem 2. Then, the following relation holds a.s.:*

$$\inf_{k,h} \limsup_{n \to \infty} \frac{1}{N} \sum_{i=1}^N l\left(y_{k,h}^i, \lambda_i, X_i\right) \leq \mathcal{V}^* \leq \sup_{k,h} \liminf_{n \to \infty} \frac{1}{N} \sum_{i=1}^N l\left(y_i, \lambda_{k,h}^i, X_i\right),$$

*where $(y_i, \lambda_i)$ are the predictions of MHA when applied on $\{H_{k,h}\}$.*

We are now ready to state and prove the optimality of MHA.

**Theorem 3** (Optimality of MHA). *Let $(y_i, \lambda_i)$ be the predictions generated by MHA when applied on $\{H_{k,h}\}$ as defined in the proof of Theorem 2. Then, for any $\gamma$-feasible process $\{X_i\}_{-\infty}^\infty$: MHA is a $\gamma$-bounded and $\gamma$-universal strategy.*

*Proof.* We first show that

$$\lim_{N \to \infty} \frac{1}{N} \sum_{i=1}^N l(y_i, \lambda_i, X_i) = \mathcal{V}^* \quad a.s. \tag{8}$$

Applying Lemma 5 in [13], we know that the $x$ updates guarantee that for every expert $H_{k,h}$,

$$\frac{1}{N} \sum_{i=1}^N l(y_i, \lambda_i, x_i) \leq \frac{1}{N} \sum_{i=1}^N l(y_{k,h}^i, \lambda_i, x_i) + \frac{C_{k,h}}{\sqrt{N}} \tag{9}$$

$$\frac{1}{N} \sum_{i=1}^N l(y_i, \lambda_i, x_i) \geq \frac{1}{N} \sum_{i=1}^N l(y_i, \lambda_{k,h}^i, x_i) - \frac{C'_{k,h}}{\sqrt{N}}, \tag{10}$$

where $C_{k,h}, C'_{k,h} > 0$ are some constants independent of $N$. In particular, using Equation (9),

$$\frac{1}{N} \sum_{i=1}^N l(y_i, \lambda_i, x_i) \leq \inf_{k,h} \left( \frac{1}{N} \sum_{i=1}^N l(y_{k,h}^i, \lambda_i, x_i) + \frac{C_{k,h}}{\sqrt{N}} \right).$$

Therefore, we get

$$\limsup_{N \to \infty} \frac{1}{N} \sum_{i=1}^{N} l(y_i, \lambda_i, x_i) \leq \limsup_{N \to \infty} \inf_{k,h} \left( \frac{1}{N} \sum_{i=1}^{N} l(y_{k,h}^i, \lambda_i, x_i) + \frac{C_{k,h}}{\sqrt{N}} \right)$$

$$\leq \inf_{k,h} \limsup_{N \to \infty} \left( \frac{1}{N} \sum_{i=1}^{N} l(y_{k,h}^i, \lambda_i, x_i) + \frac{C_{k,h}}{\sqrt{N}} \right) \leq \inf_{k,h} \limsup_{N \to \infty} \left( \frac{1}{N} \sum_{i=1}^{N} l(y_{k,h}^i, \lambda_i, x_i) \right), \quad (11)$$

where in the last inequality we used the fact that $\limsup$ is sub-additive. Using Lemma (4), we get that

$$(11) \leq \mathcal{V}^* \leq \sup_{k,h} \liminf_{n \to \infty} \frac{1}{N} \sum_{i=1}^{N} l\left(y_i, \lambda_{k,h}^i, X_i\right). \tag{12}$$

Using similar arguments and using Equation (10) we can show that

$$(12) \leq \liminf_{N \to \infty} \frac{1}{N} \sum_{i=1}^{N} l(y_i, \lambda_i, x_i). $$

Summarizing, we have

$$\limsup_{N \to \infty} \frac{1}{N} \sum_{i=1}^{N} l(y_i, \lambda_i, x_i) \leq \mathcal{V}^* \leq \liminf_{N \to \infty} \frac{1}{N} \sum_{i=1}^{N} l(y_i, \lambda_i, x_i). $$

Therefore, we can conclude that a.s. $\lim_{N \to \infty} \frac{1}{N} \sum_{i=1}^{N} l(y_i, \lambda_i, X_i) = \mathcal{V}^*$.

To show that MHA is indeed a $\gamma$-bounded strategy, we use two special experts $H_{0,0}, H_{-1,-1}$ whose predictions are $\lambda_{0,0}^n = \lambda_{\max}$ and $\lambda_{-1,-1}^n = 0$ for every $n$ and to shorten the notation, we denote

$$g(y, \lambda, x) \triangleq \lambda(c(y, x) - \gamma). $$

First, from Equation (10) applied on the expert $H_{0,0}$, we get that:

$$\limsup_{N \to \infty} \frac{1}{N} \sum_{i=1}^{N} g(y_i, \lambda_{\max}, x) \leq \limsup_{N \to \infty} \frac{1}{N} \sum_{i=1}^{N} g(y_i, \lambda_i, x). \tag{13}$$

Moreover, since $l$ is uniformly continuous, for any given $\epsilon > 0$, there exists $\delta > 0$, such that if $|\lambda' - \lambda| < \delta$, then $|l(y, \lambda', x) - l(y, \lambda, x)| < \epsilon$ for any $y \in \mathcal{Y}$ and $x \in \mathcal{X}$. We also know from the proof of Theorem 2 that $\lim_{k \to \infty} \lim_{h \to \infty} \lim_{i \to \infty} \lambda_{k,h}^i = \lambda_\infty^*$. Therefore, there exist $k_0, h_0, i_0$ such that $|\lambda_{k_0,h_0}^i - \lambda_\infty^*| < \delta$ for any $i > i_0$. Therefore,

$$\limsup_{N \to \infty} \left( \frac{1}{N} \sum_{i=1}^{N} l(y_i, \lambda_\infty^*, x_i) - \frac{1}{N} \sum_{i=1}^{N} l(y_i, \lambda_i, x_i) \right) \leq$$

$$\limsup_{N \to \infty} \left( \frac{1}{N} \sum_{i=1}^{N} l(y_i, \lambda_\infty^*, x_i) - \frac{1}{N} \sum_{i=1}^{N} l(y_i, \lambda_{k_0,h_0}^i, x_i) \right) +$$

$$\limsup_{N \to \infty} \left( \frac{1}{N} \sum_{i=1}^{N} l(y_i, \lambda_{k_0,h_0}^i, x_i) - \frac{1}{N} \sum_{i=1}^{N} l(y_i, \lambda_i, x_i) \right) \tag{14}$$

From the uniform continuity we also learn that the first summand is bounded above by $\epsilon$, and from Equation (10), we get that the last summand is bounded above by $0$. Thus,

$$(14) \leq \epsilon,$$

and since $\epsilon$ is arbitrary, we get that

$$\limsup_{N \to \infty} \left( \frac{1}{N} \sum_{i=1}^{N} l(y_i, \lambda_\infty^*, x_i) - \frac{1}{N} \sum_{i=1}^{N} l(y_i, \lambda_i, x_i) \right) \leq 0. $$

Thus, $\limsup_{N\to\infty} \frac{1}{N} \sum_{i=1}^{N} l(y_i, \lambda^*_\infty, X_i) \le \mathcal{V}^*$, and from Theorem 1 we can conclude that $\lim_{N\to\infty} \frac{1}{N} \sum_{i=1}^{N} l(y_i, \lambda^*_\infty, X_i) = \mathcal{V}^*$. Therefore, we can deduce that

$$\limsup_{N\to\infty} \frac{1}{N} \sum_{i=1}^{N} g(y_i, \lambda_i, x_i) - \limsup_{N\to\infty} \frac{1}{N} \sum_{i=1}^{N} g(y_i, \lambda^*_\infty, x_i) =$$

$$\limsup_{N\to\infty} \frac{1}{N} \sum_{i=1}^{N} g(y_i, \lambda_i, x_i) + \liminf_{N\to\infty} \frac{1}{N} \sum_{i=1}^{N} -g(y_i, \lambda^*_\infty, x_i)$$

$$\le \limsup_{N\to\infty} \frac{1}{N} \sum_{i=1}^{N} g(y_i, \lambda_i, x_i) - \frac{1}{N} \sum_{i=1}^{N} g(y_i, \lambda^*_\infty, x_i)$$

$$= \limsup_{N\to\infty} \frac{1}{N} \sum_{i=1}^{N} l(y_i, \lambda_i, x_i) - \frac{1}{N} \sum_{i=1}^{N} l(y_i, \lambda^*_\infty, x_i) = 0,$$

which results in

$$\limsup_{N\to\infty} \frac{1}{N} \sum_{i=1}^{N} g(y_i, \lambda_i, x_i) \le \limsup_{N\to\infty} \frac{1}{N} \sum_{i=1}^{N} g(y_i, \lambda^*_\infty, x_i).$$

Combining the above with Equation (13), we get that

$$\limsup_{N\to\infty} \frac{1}{N} \sum_{i=1}^{N} g(y_i, \lambda_{\max}, x_i) \le \limsup_{N\to\infty} \frac{1}{N} \sum_{i=1}^{N} g(y_i, \lambda^*_\infty, x_i).$$

Since $0 \le \lambda^*_\infty < \lambda_{\max}$, we get that MHA is $\gamma$-bounded. This also implies that

$$\limsup_{N\to\infty} \frac{1}{N} \sum_{i=1}^{N} \lambda_i (c(y_i, x_i) - \gamma) \le 0.$$

Now, if we apply Equation (10) on the expert $H_{-1,-1}$, we get that

$$\liminf_{N\to\infty} \frac{1}{N} \sum_{i=1}^{N} \lambda_i (c(y_i, x_i) - \gamma) \ge 0.$$

Thus,

$$\lim_{N\to\infty} \frac{1}{N} \sum_{i=1}^{N} \lambda_i (c(y_i, x_i) - \gamma) = 0,$$

and using Equation (8), we get that MHA is also $\gamma$-universal.

$\square$

## 5 Concluding Remarks

In this paper, we introduced the Minimax Histogram Aggregation (MHA) algorithm for multiple-objective sequential prediction. We considered the general setting where the unknown underlying process is stationary and ergodic., and given that the underlying process is $\gamma$-feasible, we extended the well-known result of [1] regarding the asymptotic lower bound of prediction with a single objective, to the case of multi-objectives. We proved that MHA is a $\gamma$-bounded strategy whose predictions also converge to the optimal solution in hindsight.

In the proofs of the theorems and lemmas above, we used the fact that the initial weights of the experts, $\alpha_{k,h}$, are strictly positive thus implying a countably infinite expert set. In practice, however, one cannot maintain an infinite set of experts. Therefore, it is customary to apply such algorithms with a finite number of experts (see [12, 9, 10]). Despite the fact that in the proof we assumed that the observation set $\mathcal{X}$ is known a priori, the algorithm can also be applied in the case that $\mathcal{X}$ is unknown by applying the doubling trick. For a further discussion on this point, see [8]. In our proofs, we relied on the compactness of the set $\mathcal{X}$. It will be interesting to see whether the universality of MHA can be sustained under unbounded processes as well. A very interesting open question would be to identify conditions allowing for finite sample bounds when predicting with multiple objectives.

## Acknowledgments

We would like to thank the anonymous reviewers for providing helpful comments. This research was supported by The Israel Science Foundation (grant No. 1890/14)

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
