[Supplementary Material · final_supp.pdf]

# Multi-Objective Non-parametric Sequential Prediction

## 1 Proofs of helping lemmas

**Lemma 2** (Continuity and Minimax). *Let $\mathcal{Y}, \Lambda, \mathcal{X}$ be compact real spaces. $l : \mathcal{Y} \times \Lambda \times \mathcal{X} \to \mathbb{R}$ be a continuous function. Denote by $\mathbb{P}(\mathcal{X})$ the space of all probability measures on $\mathcal{X}$ (equipped with the topology of weak-convergence). Then the following function $L^* : \mathbb{P}(\mathcal{X}) \to \mathbb{R}$ is continuous*

$$L^*(\mathbb{Q}) = \inf_{y \in \mathcal{Y}} \sup_{\lambda \in \Lambda} \mathbb{E}_{\mathbb{Q}} \left[ l(y, \lambda, x) \right]. \tag{1}$$

*Moreover, for any $\mathbb{Q} \in \mathbb{P}(\mathcal{X})$,*

$$\inf_{y \in \mathcal{Y}} \sup_{\lambda \in \Lambda} \mathbb{E}_{\mathbb{Q}} \left[ l(y, \lambda, x) \right] = \sup_{\lambda \in \Lambda} \inf_{y \in \mathcal{Y}} \mathbb{E}_{\mathbb{Q}} \left[ l(y, \lambda, x) \right].$$

*Proof.* $\mathcal{Y}, \Lambda, \mathcal{X}$ are compact, implying that the function $l(y, \lambda, x)$ is bounded. Therefore, the function $L : \mathcal{Y} \times \Lambda \times \mathbb{P}(\mathcal{X}) \to \mathbb{R}$, defined as

$$L(y, \lambda, \mathbb{Q}) = \mathbb{E}_{\mathbb{Q}} \left[ l(y, \lambda, x) \right], \tag{2}$$

is continuous. By applying Proposition 7.32 from [2], we have that $\sup_{\lambda \in \Lambda} \mathbb{E}_{\mathbb{Q}} \left[ l(y, \lambda, X) \right]$ is continuous in $\mathbb{Q} \times \mathcal{Y}$. Again applying the same proposition, we get the desired result. The last part of the lemma follows directly from Fan's minimax theorem [3]. $\square$

**Lemma 3** (Continuity of the optimal selection). *Let $\mathcal{Y}, \Lambda, \mathcal{X}$ be compact real spaces, and let $L$ be as defined in Equation (2). Then, there exist two measurable selection functions $h^X, h^\lambda$ such that*

$$h^y(\mathbb{Q}) \in \arg\min_{y \in \mathcal{Y}} \left( \max_{\lambda \in \Lambda} L(y, \lambda, \mathbb{Q}) \right),$$

$$h^\lambda(\mathbb{Q}) \in \arg\max_{\lambda \in \Lambda} \left( \min_{y \in \mathcal{Y}} L(y, \lambda, \mathbb{Q}) \right)$$

*for any $\mathbb{Q} \in \mathbb{P}(\mathcal{X})$. Moreover, let $L^*$ be as defined in Equation (1). Then, the set*

$$Gr(L^*) \triangleq \{ (u^*, v^*, \mathbb{Q}) \mid u^* \in h^y(\mathbb{Q}), v^* \in h^\lambda(\mathbb{Q}), \mathbb{Q} \in \mathbb{P}(\mathcal{X}) \},$$

*is closed in $\mathcal{Y} \times \Lambda \times \mathbb{P}(\mathcal{X})$.*

*Proof.* The first part of the proof follows immediately from the minimax measurable theorem of [8] due to the compactness of $\mathcal{Y}, \Lambda, \mathcal{X}$ and the properties of the loss function $L$. The proof of the second part is similar to the one presented in Theorem 3 of [1]. In order to show that $Gr(L^*)$ is closed, it is enough to show that if (i) $\mathbb{Q}_n \to \mathbb{Q}_\infty$ in $\mathbb{P}(\mathcal{X})$; (ii) $u_n \to u_\infty$ in $\mathcal{Y}$; (iii) $v_n \to v_\infty$ in $\Lambda$ and (iv) $u_n \in h^y(\mathbb{Q}_n), v_n \in h^\lambda(\mathbb{Q}_n)$ for all $n$, then,

$$u_\infty \in h^y(\mathbb{Q}_\infty), v_\infty \in h^\lambda(\mathbb{Q}_\infty).$$

The function $L(y, \lambda, \mathbb{Q})$, as defined in Equation (2), is continuous. Therefore,

$$\lim_{n \to \infty} L(u_n, v_n, \mathbb{Q}_n) = L(u_\infty, v_\infty, \mathbb{Q}_\infty).$$

It remains to show that $u_\infty \in h^y(\mathbb{Q}_\infty)$ and $v_\infty \in h^\lambda(\mathbb{Q}_\infty)$. From the optimality of $u_n$ and $v_n$, we obtain

$$L(u_\infty, v_\infty, \mathbb{Q}_\infty) = \lim_{n\to\infty} L(u_n, v_n, \mathbb{Q}_n) = \lim_{n\to\infty} L^*(\mathbb{Q}_n). \tag{3}$$

Finally, from the continuity of $L^*$ (Lemma 2), we get

$$(3) = L^*(\lim_{n\to\infty} \mathbb{Q}_n) = L^*(\mathbb{Q}_\infty),$$

which gives the desired result. $\qquad\square$

**Corollary 1.** *Under the conditions of Lemma 3. Define $L_n(y, \lambda, \mathbb{Q}) = L(y, \lambda, \mathbb{Q}) + \frac{||y||^2 - ||\lambda||^2}{n}$ and denote $h^y_{L_n}(\mathbb{Q}_n), h^\lambda_{L_n}(\mathbb{Q}_n)$ to be the measurable selection functions of $L_n$. If $\mathbb{Q}_n \to \mathbb{Q}_\infty$ weakly in $\mathbb{P}(\mathcal{X})$ and $u_n \in h^y_{L_n}(\mathbb{Q}_n), v_n \in h^\lambda_{L_n}(\mathbb{Q}_n)$, then*

$$L_n(u_n, v_n, \mathbb{Q}_n) \to L(u_\infty, v_\infty, \mathbb{Q}_\infty)$$

*almost surely for $u_\infty \in h^y(\mathbb{Q}_\infty)$ and $v_\infty \in h^\lambda(\mathbb{Q}_\infty)$.*

*Proof.* Denote $\hat{u}_n \in h^y(\mathbb{Q}_\infty)$ and $\hat{v}_n \in h^\lambda(\mathbb{Q}_\infty)$

$$|L_n(u_n, v_n, \mathbb{Q}_n) - L(u_\infty, v_\infty, \mathbb{Q}_\infty)|$$
$$\leq |L_n(u_n, v_n, \mathbb{Q}_n) - L(\hat{u}_n, \hat{v}_n, \mathbb{Q}_n)| + |L(\hat{u}_n, \hat{v}_n, \mathbb{Q}_n) - L(u_\infty, v_\infty, \mathbb{Q}_\infty)|. \tag{4}$$

Note that for every $n$ and for constant $E > 0$,

$$\min_{y\in\mathcal{Y}} \max_{\lambda\in\Lambda} L(y, \lambda, \mathbb{Q}) - \frac{||\lambda_{\max}||^2}{n} \leq \min_{y\in\mathcal{Y}} \max_{\lambda\in\Lambda} L_n(y, \lambda, \mathbb{Q})$$

$$= \min_{y\in\mathcal{Y}} \max_{\lambda\in\Lambda} \left( \mathbb{E}_\mathbb{Q}\left[l(y, \lambda, X)\right] + \frac{||y||^2 - ||\lambda||^2}{n} \right)$$

$$\leq \min_{y\in\mathcal{Y}} \max_{\lambda\in\Lambda} L(y, \lambda, \mathbb{Q}) + \frac{E}{n}.$$

Thus, for some constant $C$, $|L_n(u_n, v_n, \mathbb{Q}_n) - L(u_\infty, v_\infty, \mathbb{Q}_\infty)| < \frac{C}{n}$ and from Lemma 3, the last summand also converges to 0 as $n$ approaches $\infty$, we get the desired result, and clearly, if $h^y(\mathbb{Q}_\infty)$ and $h^\lambda(\mathbb{Q}_\infty)$ are singletons, then, the only accumulation point of $\{(v_n, u_n)\}_{n=1}^\infty$ is $(v_\infty, u_\infty)$. $\quad\square$

## 2 Proof of Theorem 1

**Theorem 1** (Optimality of $\mathcal{V}^*$)**.** *Let $\{X_i\}_{-\infty}^\infty$ be a $\gamma$-feasible process. Then, for any strategy $\mathcal{S} \in \mathcal{S}_\gamma$, the following holds a.s.*

$$\liminf_{N\to\infty} \frac{1}{N} \sum_{i=1}^N u(S(X_1^{i-1}), X_i) \geq \mathcal{V}^*.$$

*Proof.* For any given strategy $\mathcal{S} \in \mathcal{S}_\gamma$, we will look at the following sequence:

$$\frac{1}{N} \sum_{i=1}^N l(S(X_1^{i-1}), \tilde{\lambda}_i^*, X_i). \tag{5}$$

where $\tilde{\lambda}_i^* \in h^\lambda(\mathbb{P}_{X_i|X_1^{i-1}})$ Observe that

$$(5) = -\frac{1}{N} \sum_{i=1}^N \left( l(S(X_1^{i-1}), \tilde{\lambda}_i^*, X_i) - \mathbb{E}\left[l(S(X_1^{i-1}), \tilde{\lambda}_i^*, X) \mid X_1^{i-1}\right] \right)$$

$$+ \frac{1}{N} \sum_{i=1}^N \mathbb{E}\left[l(S(X_1^{i-1}), \tilde{\lambda}_i^*, X_i) \mid X_1^{i-1}\right].$$

Since $A_i = l(S(X_1^{i-1}), \tilde{\lambda}_i^*, X_i) - \mathbb{E}\left[l(S(X_1^{i-1}), \tilde{\lambda}_i^*, X_i) \mid X_1^{i-1}\right]$ is a martingale difference sequence, the last summand converges to 0 a.s., by the strong law of large numbers (see, e.g., [9]). Therefore,

$$\liminf_{N \to \infty} \frac{1}{N} \sum_{i=1}^{N} l(S(X_1^{i-1}), \tilde{\lambda}_i^*, X_i) = \liminf_{N \to \infty} \frac{1}{N} \sum_{i=1}^{N} \mathbb{E}\left[l(S(X_1^{i-1}), \tilde{\lambda}_i^*, X_i) \mid X_1^{i-1}\right]$$

$$\geq \liminf_{N \to \infty} \frac{1}{N} \sum_{i=1}^{N} \min_{y \in \mathcal{Y}()} \mathbb{E}\left[l(y, \tilde{\lambda}_i^*, X_i) \mid X_1^{i-1}\right], \tag{6}$$

where the minimum is taken w.r.t. all the $\sigma(X_1^{i-1})$-measurable functions. Because the process is stationary, we get for $\hat{\lambda}_i^* \in h^\lambda(\mathbb{P}_{X_0 | X_{1-i}^{-1}})$,

$$(6) = \liminf_{N \to \infty} \frac{1}{N} \sum_{i=1}^{N} \min_{y \in \mathcal{Y}()} \mathbb{E}\left[l(y, \hat{\lambda}_i^*, X_0) \mid X_{1-i}^{-1}\right] \tag{7}$$

$$= \liminf_{N \to \infty} \frac{1}{N} \sum_{i=1}^{N} L^*(\mathbb{P}_{X_0 | X_{1-i}^{-1}}). \tag{8}$$

Using Levy's zero-one law, $\mathbb{P}_{X_0 | X_{1-i}^{-1}} \to \mathbb{P}_\infty$ weakly as $i$ approaches $\infty$ and from Lemma 2 we know that $L^*$ is continuous. Therefore, we can apply Lemma 1 and get that a.s.

$$(8) = \mathbb{E}\left[L^*(\mathbb{P}_\infty)\right] = \mathbb{E}\left[\mathbb{E}_{\mathbb{P}_\infty}\left[l\left(y_\infty^*, \lambda_\infty^*, X_0\right)\right]\right] = \mathbb{E}\left[\mathcal{L}\left(y_\infty^*, \lambda_\infty^*, X_0\right)\right]. \tag{9}$$

Note also, that due to the complementary slackness condition of the optimal solution, i.e., $\lambda_\infty^*(\mathbb{E}_{\mathbb{P}_\infty}\left[c(y_\infty^*, X_0)\right] - \gamma) = 0$, we get

$$(9) = \mathbb{E}\left[\mathbb{E}_{\mathbb{P}_\infty}\left[u\left(y_\infty^*, X_0\right)\right]\right] = \mathcal{V}^*.$$

From the uniqueness of $\lambda_\infty^*$, and using Lemma 3 $\hat{\lambda}_i^* \to \lambda_\infty^*$ as $i$ approaches $\infty$. Moreover, since $l$ is continuous on a compact set, $l$ is also uniformly continuous. Therefore, for any given $\epsilon > 0$, there exists $\delta > 0$, such that if $|\lambda' - \lambda| < \delta$, then

$$|l(y, \lambda', x) - l(y, \lambda, x)| < \epsilon$$

for any $y \in \mathcal{Y}$ and $x \in \mathcal{X}$. Therefore, there exists $i_0$ such that if $i > i_0$ then $|l(y, \hat{\lambda}_i^*, x) - l(y, \lambda_\infty^*, x)| < \epsilon$ for any $y \in \mathcal{Y}$ and $x \in \mathcal{X}$. Thus,

$$\liminf_{N \to \infty} \frac{1}{N} \sum_{i=1}^{N} l(S(X_1^{i-1}), \lambda_\infty^*, X_i) - \liminf_{N \to \infty} \frac{1}{N} \sum_{i=1}^{N} l(S(X_1^{i-1}), \hat{\lambda}_i^*, X_i)$$

$$= \liminf_{N \to \infty} \frac{1}{N} \sum_{i=1}^{N} l(S(X_1^{i-1}), \lambda_\infty^*, X_i) + \limsup_{N \to \infty} \frac{1}{N} \sum_{i=1}^{N} -l(S(X_1^{i-1}), \hat{\lambda}_i^*, X_i)$$

$$\geq \liminf_{N \to \infty} \frac{1}{N} \sum_{i=1}^{N} l(S(X_1^{i-1}), \hat{\lambda}_i^*, X_i) - \frac{1}{N} \sum_{i=1}^{N} l(S(X_1^{i-1}), \lambda_\infty^*, X_i) \geq -\epsilon \ \ a.s.,$$

and since $\epsilon$ is arbitrary,

$$\liminf_{N \to \infty} \frac{1}{N} \sum_{i=1}^{N} l(S(X_1^{i-1}), \lambda_\infty^*, X_i) \geq \liminf_{N \to \infty} \frac{1}{N} \sum_{i=1}^{N} l(S(X_1^{i-1}), \hat{\lambda}_i^*, X_i).$$

Therefore we can conclude that

$$\liminf_{N \to \infty} \frac{1}{N} \sum_{i=1}^{N} l(S(X_1^{i-1}), \lambda_\infty^*, X_i) \geq \mathcal{V}^* \ \ a.s.$$

We finish the proof by noticing that since $\mathcal{S} \in \mathcal{S}_\gamma$, then by definition

$$\limsup_{N \to \infty} \frac{1}{N} \sum_{i=1}^{N} c(S(X_1^{i-1}), X_i) \leq \gamma \ \ a.s.$$

and since $\lambda_\infty^*$ is non negative, we will get the desired result. $\qquad\square$

# 3 Proof of Theorem 2

**Theorem 2.** *Assume that $\{X_i\}_{-\infty}^{\infty}$ is a $\gamma$-feasible process. Then, it is possible to construct a countable set of experts $\{H_{k,h}\}$ for which*

$$\lim_{k\to\infty}\lim_{h\to\infty}\lim_{n\to\infty}\frac{1}{N}\sum_{i=1}^{N}l(y_{k,h}^{i},\lambda_{k,h}^{i},X_i)=\mathcal{V}^{*}\ a.s.,$$

*where $(y_{k,h}^{i},\lambda_{k,h}^{i})$ are the predictions made by expert $H_{k,h}$ at round $i$.*

*Proof.* We start by defining a countable set of experts $\{H_{k,h}\}$ as follow: For $h=1,2,\dots$, let $P_h=\{A_{h,j}\mid j=1,2,...,m_h\}$ be a sequence of finite partitions of $\mathcal{X}$ such that: (i) any cell of $P_{h+1}$ is a subset of a cell of $P_h$ for any $h$. Namely, $P_{h+1}$ is a refinement of $P_h$; (ii) for a set $A$, if $diam(A)=\sup_{x,y\in A}||x-y||$ denotes the diameter of $A$, then for any sphere $B$ centered at the origin,

$$\lim_{h\to\infty}\max_{j:A_{h,j}\cap B\neq\emptyset}diam(A_{h,j})=0.$$

Define the corresponding quantizer $q_h(x)=j$, if $x\in A_{h,j}$. Thus, for any $n$ and $X_1^n$, we define $Q_h(X_1^n)$ as the sequence $q_h(x_1),\dots,q_h(x_n)$. For expert $H_{k,h}$, we define for $k>0$, a $k$-long string of positive integers, denoted by $w$, the following set,

$$B_{k,h}^{w,(1,n-1)}\triangleq\{x_i\mid k<i<n,\ Q_h(X_{i-k}^{i-1})=w\}.$$

We define also

$$h_{k,h}^{y}(X_1^{n-1},w)\triangleq\arg\min_{y\in\mathcal{Y}}\left(\max_{\lambda\in\Lambda}\frac{1}{|B_{k,h}^{w,(1,n-1)}|}\sum_{x_i\in B_{k,h}^{w,(1,n-1)}}l_{k,h,n}(y,\lambda,x_i)\right)$$

$$h_{k,h}^{\lambda}(X_1^{n-1},w)\triangleq\arg\max_{\lambda\in\Lambda}\left(\min_{y\in\mathcal{Y}}\frac{1}{|B_{k,h}^{w,(1,n-1)}|}\sum_{x_i\in B_{k,h}^{w,(1,n-1)}}l_{k,h,n}(y,\lambda,x_i)\right)$$

for

$$l_{k,h,n}(y,\lambda,x)\triangleq l(y,\lambda,x)+\left(||y||^2-||\lambda||^2\right)\left(\frac{1}{n}+\frac{1}{h}+\frac{1}{k}\right)$$

and we will set $h_{k,h}^{y}(X_1^{n-1},w)=y_0$ and $h_{k,h}^{\lambda}(X_1^{n-1},w)=\lambda_0$ for arbitrary $(y_0,\lambda_0)\in\mathcal{Y}\times\Lambda$ if $B_{k,h}^{w,(1,n-1)}$ is empty. Using the above, we define the predictions of $H_{k,h}$ to be:

$$H_{k,h}^{y}(X_1^{n-1})=h_{k,h}^{y}(X_1^{n-1},Q(X_{n-k}^{n-1})),\ n=1,2,3....$$
$$H_{k,h}^{\lambda}(X_1^{n-1})=h_{k,h}^{\lambda}(X_1^{n-1},Q(X_{n-k}^{n-1})),\ n=1,2,3....$$

We will add two experts: $H_{0,0}$ whose predictions are always $(y_0,\lambda_{\max})$ and $H_{-1,-1}$ whose predictions are always $(y_0,0)$.

Fixing $k,h>0$ and $w$, we will define a (random) measure $\mathbb{P}_{j,w}^{(k.h)}$ that is the measure concentrated on the set $B_{k,h}^{w,(0,1-j)}$, defined by

$$\mathbb{P}_{j,w}^{(k,h)}(A)=\frac{\sum_{X_i\in B_{k,h}^{w,(0,1-j)}}1_A(X_i)}{|B_{k,h}^{w,(0,1-j)}|},$$

where $1_A$ denotes the indicator function of the set $A\subset\mathcal{X}$. If the above set $B_{k,h}^{w}$ is empty, then let $P_{j,w}^{(k,h)}(A)=\delta(x')$ be the probability measure concentrated on arbitrary vector $x'\in\mathcal{X}$.

In other words, $\mathbb{P}_{j,w}^{(k,h)}(A)$ is the relative frequency of the the vectors among $X_{1-j+k}, \ldots, X_0$ that fall in the set $A$. Applying Lemma 1 twice, it is straightforward to prove that for all $w$, w.p. 1

$$\mathbb{P}_{j,w}^{(k,h)} \rightarrow \begin{cases} \mathbb{P}_{X_0|G_l(X_{-k}^{-1})=w} & \mathbb{P}(G_l(X_{-k}^{-1}) = w) > 0 \\ \delta(x') & otherwise \end{cases}$$

weakly as $j \rightarrow \infty$, where $\mathbb{P}_{X_0|G_l(X_{-k}^{-1})=w}$ denotes the distribution of the vector $X_0$ conditioned on the event $G_l(X_{-k}^{-1}) = w$. To see this, let $f$ be a bounded continuous function. Then,

$$\int f(x)\mathbb{P}_{j,w}^{(k,h)}(dx) = \frac{\frac{1}{|1-j+k|} \sum_{X_i \in B_{k,h}^{w,(0,1-j)}} f(X_i)}{\frac{1}{|1-j+k|}|B_{k,h}^{w,(0,1-j)}|}$$

$$\rightarrow \frac{\mathbb{E}\left[f(X_0)1_{G_l(X_{-k}^{-1})=w}(X_0)\right]}{\mathbb{P}(G_l(X_{-k}^{-1}) = w)} = \mathbb{E}\left[f(X_0) \mid G_l(X_{-k}^{-1}) = w\right],$$

and in case $\mathbb{P}(||X_{-k}^{-1} - s|| \leq c/l) = 0$, then w.p. 1, $\mathbb{P}_{j,w}^{(k,h)}$ is concentrated on $x'$ for all $j$. We will denote the limit distribution of $\mathbb{P}_{j,w}^{(k,h)}$ by $\mathbb{P}_w^{*(k,h)}$.

By definition, $\left(h_{k,h}^y(X_{1-n}^{-1}, w), h_{k,h}^\lambda(X_{1-n}^{-1}, w)\right)$ is the minimax of $l_{n,k,h}$ w.r.t. $\mathbb{P}_{j,w}^{(k,h)}$. The sequence of functions $l_{n,k,h}$ converges uniformly as $n$ approaches $\infty$ to

$$l_{k,h}(y, \lambda, x) = l(y, \lambda, x) + \left(||y||^2 - ||\lambda||^2\right)\left(\frac{1}{h} + \frac{1}{k}\right).$$

Note also that for any fixed $\mathbb{Q}$, $L_{k,h}(y, \lambda, \mathbb{Q}) = \mathbb{E}_\mathbb{Q}\left[l_{k,h}(y, \lambda, X)\right]$ is strictly convex in $y$ and strictly concave in $\lambda$, and therefore, has a unique saddle-point (see, e.g., [7]). Therefore, since $w$ is arbitrary, and following a Corollary 1 of Lemma 3, we get that a.s.

$$y_{k,h}^n \rightarrow y_{k,h}^* \qquad \lambda_{k,h}^n \rightarrow \lambda_{k,h}^*,$$

where $\left(y_{k,h}^*, \lambda_{k,h}^*\right)$ is the minimax of $L_{k,h}$ w.r.t. $\mathbb{P}_{X_{-k}^{-1}}^{*(k,h)}$. Thus, we can apply Lemma 1 and conclude that as $N$ approaches $\infty$,

$$\frac{1}{N} \sum_{i=1}^N l(y_{k,h}^i, \lambda_{k,h}^i, X_i) \rightarrow \mathbb{E}\left[l(y_{k,h}^*, \lambda_{k,h}^*, X_0)\right].$$

a.s.. We now evaluate

$$\lim_{h\to\infty} \mathbb{E}\left[l(y_{k,h}^*, \lambda_{k,h}^*, X_0)\right].$$

Using the properties of the partition $P_h$ (see, e.g., [4, 5]), we get that

$$\mathbb{P}_{X_{-k}^{-1}}^{*(k,h)} \rightarrow \mathbb{P}_{\left\{X_0|X_{-k}^{-1}\right\}}$$

weakly as $h \rightarrow \infty$. Moreover, the sequence of functions $l_{k,h}$ converges uniformly as $h$ approaches $\infty$

$$l_k(y, \lambda, x) = l(y, \lambda, x) + \frac{||y||^2 - ||\lambda||^2}{k}.$$

Note also, that for any fixed $\mathbb{Q}$, $L_k(y, \lambda, \mathbb{Q}) = \mathbb{E}_\mathbb{Q}\left[l_k(y, \lambda, X)\right]$ is strictly convex-concave, and therefore, has a unique saddle point. Accordingly, by applying Corollary 1 again, we get that a.s.

$$y_{k,h}^* \rightarrow y_k^* \qquad \lambda_{k,h}^* \rightarrow \lambda_k^*,$$

where $(y_k^*, \lambda_k^*)$ is the minimax of $L_k$ w.r.t. $\mathbb{P}_{\left\{X_0|X_{-k}^{-1}\right\}}$. Therefore, as $h$ approaches $\infty$,

$$l(y_{k,h}^*, \lambda_{k,h}^*, X_0) \rightarrow l(y_k^*, \lambda_k^*, X_0)$$

a.s.. Thus, by Lebesgue's dominated convergence,

$$\lim_{h\to\infty} \mathbb{E}\left[l(y_{k,h}^*, \lambda_{k,h}^*, X_0)\right] = \mathbb{E}\left[l(y_k^*, \lambda_k^*, X_0)\right].$$

Notice that for any $\mathbb{Q} \in \mathbb{P}(\mathcal{X})$, the distance between the saddle point of $L_k$ w.r.t. $\mathbb{Q}$ and the the saddle point of $L$ w.r.t. $\mathbb{Q}$ converges to 0 as $k$ approaches $\infty$. To see this, notice that

$$\min_{y \in \mathcal{Y}} \max_{\lambda \in \Lambda} L(y, \lambda, \mathbb{Q}) - \frac{||\lambda_{\max}||^2}{k} \leq \min_{y \in \mathcal{Y}} \max_{\lambda \in \Lambda} L_k(y, \lambda, \mathbb{Q})$$

$$\leq \min_{y \in \mathcal{Y}} \max_{\lambda \in \Lambda} L(y, \lambda, \mathbb{Q}) + \frac{E}{k}$$

for some constant $E$, since $\mathcal{Y}$ is bounded. The last part in our proof will be to show that if $(\hat{y}_k^*, \hat{\lambda}_k^*)$ is the minimax of $L$ w.r.t. $\mathbb{P}_{\{X_0 | X_{-k}^{-1}\}}$, then as $k$ approaches $\infty$, $\mathbb{E}\left[l\left(\hat{y}_k^*, \hat{\lambda}_k^*, X_0\right)\right]$ will converge a.s. to $\mathcal{V}^*$ and so $\mathbb{E}\left[l\left(y_k^*, \lambda_k^*, X_0\right)\right]$.

To show this, we will use the sub-martingale convergence theorem twice. First, we define $Z_k$ as

$$Z_k \triangleq \min_{y \in \mathcal{Y}()} \mathbb{E}\left[\max_{\lambda \in \Lambda()} \mathbb{E}\left[l\left(y, \lambda, X_0\right) \mid X_{-\infty}^{-1}\right] \mid X_{-k}^{-1}\right],$$

where the minimum is taken w.r.t. all $\sigma(X_{-k}^{-1})$-measurable strategies and the maximum is taken w.r.t. all $\sigma(X_{-\infty}^{-1})$-measurable strategies. Notice that $Z_k$ is a super-martingale. We can see this by using the tower property of conditional expectations,

$$\mathbb{E}[Z_{k+1} \mid X_{-k}^{-1}] = \mathbb{E}\left[\mathbb{E}\left[Z_{k+1} \mid X_{-k-1}^{-1}\right] \mid X_{-k}^{-1}\right]$$

and since $Z_{k+1}$ is the optimal choice in $\mathcal{Y}$ w.r.t. to $X_{-k-1}^{-1}$,

$$\leq \mathbb{E}\left[\mathbb{E}[Z_k \mid X_{-k-1}^{-1}] \mid X_{-k}^{-1}\right] = \mathbb{E}[Z_k \mid X_{-k}^{-1}] = Z_k.$$

Note also that $\mathbb{E}[Z_k]$ is uniformly bounded. Therefore, we can apply the super-martingale convergence theorem and get that $Z_k \to Z_\infty$ a.s., where,

$$Z_\infty = \mathbb{E}\left[l(y_\infty^*, \lambda_\infty^*, X_0) \mid X_{-\infty}^{-1}\right] = \mathcal{V}^*,$$

and by using Lebesgue's dominated convergence theorem, also $\mathbb{E}[Z_k] \to \mathbb{E}[Z_\infty] = \mathcal{V}^*$. Using the same arguments, $Z_k'$, defined as

$$Z_k' \triangleq \max_{\lambda \in \Lambda()} \mathbb{E}\left[\min_{y \in \mathcal{Y}()} \mathbb{E}\left[l\left(y, \lambda, X_0\right) \mid X_{-\infty}^{-1}\right] \mid X_{-k}^{-1}\right],$$

where the maximum is taken w.r.t. all $\sigma(X_{-k}^{-1})$-measurable strategies and the minimum is taken w.r.t. all $\sigma(X_{-\infty}^{-1})$-measurable strategies, is a sub-martingale that also converges a.s. to $Z_\infty$ and thus $\mathbb{E}[Z_k'] \to \mathbb{E}[Z_\infty] = \mathcal{V}^*$.

We conclude the proof by noticing that the following relation holds for any $k$,

$$\mathbb{E}[Z_k'] = \mathbb{E}\left[\max_{\lambda \in \Lambda()} \mathbb{E}\left[\min_{y \in \mathcal{Y}()} \mathbb{E}\left[l\left(y, \lambda, X_0\right) \mid X_{-\infty}^{-1}\right] \mid X_{-k}^{-1}\right]\right]$$

$$\leq \mathbb{E}\left[\max_{\lambda \in \Lambda()} \mathbb{E}\left[\mathbb{E}\left[l\left(\hat{y}_k^*, \lambda, X_0\right) \mid X_{-\infty}^{-1}\right] \mid X_{-k}^{-1}\right]\right]$$

$$= \mathbb{E}\left[\max_{\lambda \in \Lambda()} \mathbb{E}\left[l\left(\hat{y}_k^*, \lambda, X_0\right) \mid X_{-k}^{-1}\right]\right] = \mathbb{E}\left[l\left(\hat{y}_k^*, \hat{\lambda}_k^*, X_0\right)\right],$$

and using similar arguments we can show that also

$$\mathbb{E}\left[l\left(\hat{y}_k^*, \hat{\lambda}_k^*, X_0\right)\right] \leq \mathbb{E}[Z_k],$$

and since both $\mathbb{E}[Z_k]$ and $\mathbb{E}[Z_k']$ converge to $\mathcal{V}^*$, we get the desired result.

$\square$

# 4 Proof of Theorem 3

Before proving the main theorem regarding MHA, we now state and prove the following lemma, which is used in the proof of the main result regarding MHA.

**Lemma 4.** *Let $\{H_{k,h}\}$ be a countable set of experts as defined in the proof of Theorem 2. Then, the following relation holds a.s.:*

$$\inf_{k,h} \limsup_{n\to\infty} \frac{1}{N} \sum_{i=1}^{N} l\left(y_{k,h}^i, \lambda_i, X_i\right) \leq \mathcal{V}^*$$

$$\leq \sup_{k,h} \liminf_{n\to\infty} \frac{1}{N} \sum_{i=1}^{N} l\left(y_i, \lambda_{k,h}^i, X_i\right),$$

*where $(y_i, \lambda_i)$ are the predictions of MHA when applied on $\{H_{k,h}\}$.*

*Proof.* Set

$$f(y, \mathbb{Q}) \triangleq \max_{\lambda \in \Lambda} \mathbb{E}_{\mathbb{Q}}\left[l\left(y, \lambda, X_0\right)\right].$$

We will start from the LHS,

$$\inf_{k,h} \limsup_{n\to\infty} \frac{1}{N} \sum_{i=1}^{N} l\left(y_{k,h}^i, \lambda_i, X_i\right), \tag{10}$$

and similarly to Lemma 1, by using the strong law of large numbers we can write

$$(10) = \inf_{k,h} \limsup_{n\to\infty} \frac{1}{N} \sum_{i=1}^{N} \mathbb{E}\left[l\left(y_{k,h}^i, \lambda_i, X_0\right) \mid X_{1-i}^{-1}\right]$$

$$\leq \inf_{k,h} \limsup_{n\to\infty} \frac{1}{N} \sum_{i=1}^{N} f(y_{k,h}^i, \mathbb{P}_{X_0|X_{1-i}^{-1}}) \quad a.s. \tag{11}$$

For fixed $k, h > 0$, from the proof of Theorem (2), $y_{k,h}^i \to y_{k,h}^*$ a.s. as $i$ approaches $\infty$, and from Levy's zero-one law also $\mathbb{P}_{X_0|X_{1-i}^{-1}} \to \mathbb{P}_\infty$ weakly. From Lemma 2 we know that $f$ is continuous, therefore, we can apply Lemma 1 and get that

$$(11) = \inf_{k,h} \mathbb{E}\left[\mathbb{E}\left[f(y_{k,h}^*, \mathbb{P}_\infty)\right]\right] \leq \lim_{k\to\infty} \lim_{l\to\infty} \mathbb{E}\left[f(y_{k,h}^*, \mathbb{P}_\infty)\right]. \tag{12}$$

From the uniqueness of the saddle point and from the proof of Theorem (2), for fiked $k > 0$,

$$\lim_{h\to\infty} y_{k,h}^* \to y_k^*$$

a.s.. Thus, from the continuity of $f$ we get that

$$\lim_{h\to\infty} f(y_{k,h}^*, \mathbb{P}_\infty) \to f(y_k^*, \mathbb{P}_\infty)$$

and again by Lebesgue's dominated convergence,

$$(12) = \lim_{k\to\infty} \mathbb{E}\left[f(y_k^*, \mathbb{P}_\infty)\right] = \lim_{k\to\infty} \mathbb{E}\left[\max_{\lambda \in \Lambda} \mathbb{E}_{\mathbb{P}_\infty}\left[l\left(y_k^*, \lambda, X_0\right)\right]\right]. \tag{13}$$

Now, from Theorem 2 we know that every accumulation point of the sequence $\{y_k^*\}$ is in the optimal set

$$\arg\min_{y\in\mathcal{Y}}\left(\max_{\lambda\in\Lambda} \mathbb{E}_{\mathbb{P}_\infty}\left[l\left(y, \lambda, X_0\right)\right]\right).$$

Therefore a.s.

$$\lim_{k\to\infty} \max_{\lambda\in\Lambda} \mathbb{E}_{\mathbb{P}_\infty}\left[l\left(y_k^*, \lambda, X_0\right)\right] \to \mathbb{E}_{\mathbb{P}_\infty}\left[l\left(y_\infty^*, \lambda_\infty^*, X_0\right)\right],$$

and using Lebesgue's dominated convergence,

$$(13) = \mathbb{E}\left[\mathbb{E}_{\mathbb{P}_\infty}\left[l\left(y_\infty^*, \lambda_\infty^*, X_0\right)\right]\right] = \mathcal{V}^*.$$

Using similar arguments, we can show the second part of the lemma.

$\square$

We are now ready to state and prove the optimality of MHA.

**Theorem 3** (Optimality of MHA). *Let $(y_i, \lambda_i)$ be the predictions generated by MHA when applied on $\{H_{k,h}\}$ as defined in the proof of Theorem 2. Then, for any $\gamma$-feasible process $\{X_i\}_{-\infty}^{\infty}$: MHA is a $\gamma$-bounded and $\gamma$-universal strategy.*

*Proof.* We first show that

$$\lim_{N \to \infty} \frac{1}{N} \sum_{i=1}^{N} l(y_i, \lambda_i, X_i) = \mathcal{V}^* \quad a.s. \tag{14}$$

Applying Lemma 5 in [6], we know that the $x$ updates guarantee that for every expert $H_{k,h}$,

$$\frac{1}{N} \sum_{i=1}^{N} l(y_i, \lambda_i, x_i) \leq \frac{1}{N} \sum_{i=1}^{N} l(y_{k,h}^i, \lambda_i, x_i) + \frac{C_{k,h}}{\sqrt{N}} \tag{15}$$

$$\frac{1}{N} \sum_{i=1}^{N} l(y_i, \lambda_i, x_i) \geq \frac{1}{N} \sum_{i=1}^{N} l(y_i, \lambda_{k,h}^i, x_i) - \frac{C'_{k,h}}{\sqrt{N}}, \tag{16}$$

where $C_{k,h}, C'_{k,h} > 0$ are some constants independent of $N$. In particular, using Equation (15),

$$\frac{1}{N} \sum_{i=1}^{N} l(y_i, \lambda_i, x_i) \leq \inf_{k,h} \left( \frac{1}{N} \sum_{i=1}^{N} l(y_{k,h}^i, \lambda_i, x_i) + \frac{C_{k,h}}{\sqrt{N}} \right).$$

Therefore, we get

$$\limsup_{N \to \infty} \frac{1}{N} \sum_{i=1}^{N} l(y_i, \lambda_i, x_i)$$

$$\leq \limsup_{N \to \infty} \inf_{k,h} \left( \frac{1}{N} \sum_{i=1}^{N} l(y_{k,h}^i, \lambda_i, x_i) + \frac{C_{k,h}}{\sqrt{N}} \right)$$

$$\leq \inf_{k,h} \limsup_{N \to \infty} \left( \frac{1}{N} \sum_{i=1}^{N} l(y_{k,h}^i, \lambda_i, x_i) + \frac{C_{k,h}}{\sqrt{N}} \right)$$

$$\leq \inf_{k,h} \limsup_{N \to \infty} \left( \frac{1}{N} \sum_{i=1}^{N} l(y_{k,h}^i, \lambda_i, x_i) \right), \tag{17}$$

where in the last inequality we used the fact that $\limsup$ is sub-additive. Using Lemma (4), we get that

$$(17) \leq \mathcal{V}^*$$

$$\leq \sup_{k,h} \liminf_{n \to \infty} \frac{1}{N} \sum_{i=1}^{N} l\left(y_i, \lambda_{k,h}^i, X_i\right). \tag{18}$$

Using similar arguments and using Equation (16) we can show that

$$(18) \leq \liminf_{N \to \infty} \frac{1}{N} \sum_{i=1}^{N} l(y_i, \lambda_i, x_i).$$

Summarizing, we have

$$\limsup_{N \to \infty} \frac{1}{N} \sum_{i=1}^{N} l(y_i, \lambda_i, x_i) \leq \mathcal{V}^* \leq \liminf_{N \to \infty} \frac{1}{N} \sum_{i=1}^{N} l(y_i, \lambda_i, x_i).$$

Therefore, we can conclude that a.s.

$$\lim_{N \to \infty} \frac{1}{N} \sum_{i=1}^{N} l(y_i, \lambda_i, X_i) = \mathcal{V}^*.$$

To show that MHA is indeed a $\gamma$-bounded strategy and to shorten the notation, we will denote

$$g(y, \lambda, x) \triangleq \lambda(c(y, x) - \gamma).$$

First, from Equation (16) applied on the expert $H_{0,0}$, we get that:

$$\limsup_{N \to \infty} \frac{1}{N} \sum_{i=1}^{N} g(y_i, \lambda_{\max}, x) \le \limsup_{N \to \infty} \frac{1}{N} \sum_{i=1}^{N} g(y_i, \lambda_i, x). \tag{19}$$

Moreover, since $l$ is uniformly continuous, for any given $\epsilon > 0$, there exists $\delta > 0$, such that if $|\lambda' - \lambda| < \delta$, then

$$|l(y, \lambda', x) - l(y, \lambda, x)| < \epsilon$$

for any $y \in \mathcal{Y}$ and $x \in \mathcal{X}$. We also know that

$$\lim_{k \to \infty} \lim_{h \to \infty} \lim_{i \to \infty} \lambda_{k,h}^{i} = \lambda_{\infty}^{*}.$$

Therefore, there exist $k_0, h_0, i_0$ such that $|\lambda_{k_0,h_0}^{i} - \lambda_{\infty}^{*}| < \delta$ for any $i > i_0$. Since $\lim_{k \to \infty} \lambda_k^* = \lambda_{\infty}^*$ there exists $k_0$ such that $|\lambda_{k_0}^* - \lambda_{\infty}^*| < \frac{\delta}{3}$. Note that $\lim_{h \to \infty} \lambda_{k_0,h}^* = \lambda_{k_0}^*$, so there exists $h_0$ such that $|\lambda_{k_0,h_0}^* - \lambda_{k_0}^*| < \frac{\delta}{3}$. Finally, since $\lim_{i \to \infty} \lambda_{k_0,l_0}^{i} = \lambda_{k_0,l_0}^*$, there exists $i_0$ such that if $i > i_0$, then $|\lambda_{k_0,l_0}^{i} - \lambda_{k_0,l_0}^*| < \frac{\delta}{3}$. Combining all the above, we get that for $k_0, h_0, i_0$ if $i > i_0$, then

$$|\lambda_{k_0,h_0}^{i} - \lambda_{\infty}^{*}| < |\lambda_{k_0,h_0}^{i} - \lambda_{k_0,h_0}^{*}| + |\lambda_{k_0,h_0}^{*} - \lambda_{k_0}^{*}| + |\lambda_{k_0}^{*} - \lambda_{\infty}^{*}| < \delta.$$

Therefore,

$$\limsup_{N \to \infty} \left( \frac{1}{N} \sum_{i=1}^{N} l(y_i, \lambda_{\infty}^*, x_i) - \frac{1}{N} \sum_{i=1}^{N} l(y_i, \lambda_i, x_i) \right) \le$$

$$\limsup_{N \to \infty} \left( \frac{1}{N} \sum_{i=1}^{N} l(y_i, \lambda_{\infty}^*, x_i) - \frac{1}{N} \sum_{i=1}^{N} l(y_i, \lambda_{k_0,h_0}^{i}, x_i) \right) +$$

$$\limsup_{N \to \infty} \left( \frac{1}{N} \sum_{i=1}^{N} l(y_i, \lambda_{k_0,h_0}^{i}, x_i) - \frac{1}{N} \sum_{i=1}^{N} l(y_i, \lambda_i, x_i) \right) \tag{20}$$

From the uniform continuity we also learn that the first summand is bounded above by $\epsilon$, and from Equation (16), we get that the last summand is bounded above by 0. Thus,

$$(20) \le \epsilon,$$

and since $\epsilon$ is arbitrary, we get that

$$\limsup_{N \to \infty} \left( \frac{1}{N} \sum_{i=1}^{N} l(y_i, \lambda_{\infty}^*, x_i) - \frac{1}{N} \sum_{i=1}^{N} l(y_i, \lambda_i, x_i) \right) \le 0.$$

Thus,

$$\limsup_{N \to \infty} \frac{1}{N} \sum_{i=1}^{N} l(y_i, \lambda_{\infty}^*, X_i) \le \mathcal{V}^*,$$

and from Theorem 1 we can conclude that

$$\lim_{N \to \infty} \frac{1}{N} \sum_{i=1}^{N} l(y_i, \lambda_{\infty}^*, X_i) = \mathcal{V}^*.$$

Therefore, we can deduce that

$$\limsup_{N\to\infty} \frac{1}{N}\sum_{i=1}^{N} g(y_i, \lambda_i, x_i) - \limsup_{N\to\infty} \frac{1}{N}\sum_{i=1}^{N} g(y_i, \lambda_\infty^*, x_i) =$$

$$\limsup_{N\to\infty} \frac{1}{N}\sum_{i=1}^{N} g(y_i, \lambda_i, x_i) + \liminf_{N\to\infty} \frac{1}{N}\sum_{i=1}^{N} -g(y_i, \lambda_\infty^*, x_i)$$

$$\leq \limsup_{N\to\infty} \frac{1}{N}\sum_{i=1}^{N} g(y_i, \lambda_i, x_i) - \frac{1}{N}\sum_{i=1}^{N} g(y_i, \lambda_\infty^*, x_i)$$

$$= \limsup_{N\to\infty} \frac{1}{N}\sum_{i=1}^{N} l(y_i, \lambda_i, x_i) - \frac{1}{N}\sum_{i=1}^{N} l(y_i, \lambda_\infty^*, x_i) = 0,$$

which results in

$$\limsup_{N\to\infty} \frac{1}{N}\sum_{i=1}^{N} g(y_i, \lambda_i, x_i) \leq \limsup_{N\to\infty} \frac{1}{N}\sum_{i=1}^{N} g(y_i, \lambda_\infty^*, x_i).$$

Combining the above with Equation (19), we get that

$$\limsup_{N\to\infty} \frac{1}{N}\sum_{i=1}^{N} g(y_i, \lambda_{\max}, x_i)$$

$$\leq \limsup_{N\to\infty} \frac{1}{N}\sum_{i=1}^{N} g(y_i, \lambda_\infty^*, x_i).$$

Since $0 \leq \lambda_\infty^* < \lambda_{\max}$, we get that MHA is $\gamma$-bounded. This also implies that

$$\limsup_{N\to\infty} \frac{1}{N}\sum_{i=1}^{N} \lambda_i(c(y_i, x_i) - \gamma) \leq 0.$$

Now, if we apply Equation (16) on the expert $H_{-1,-1}$, we get that

$$\liminf_{N\to\infty} \frac{1}{N}\sum_{i=1}^{N} \lambda_i(c(y_i, x_i) - \gamma) \geq 0.$$

Thus,

$$\lim_{N\to\infty} \frac{1}{N}\sum_{i=1}^{N} \lambda_i(c(y_i, x_i) - \gamma) = 0,$$

and using Equation (14), we get that MHA is also $\gamma$-universal.

$\square$