[Reviews · NeurIPS 2017]

Reviewer 1



This paper presents an asymptotic analysis for nonparametric sequential prediction when there are multiple objectives. While the paper has some relevance to a certain subset of the machine learning community, I have some concerns about the paper's relevance to NIPS, and I also am unclear on some of the basic setup of the paper. I discuss these issues in the remainder of the review. Overall, I think the results in the paper are technically strong and, with the right motivation for $\gamma$-feasibility (see below) I would be weakly positive on the paper. Relevance: While I can see that the contents of the paper have some relevance to machine learning, I feel that this sort of paper would fit much better in a conference like COLT or ALT, given the technical level and given how much this paper can benefit from the additional space offered by those venues. I am not convinced that the problem addressed by this paper is interesting to the broader machine learning community and even many theoretical researchers within online learning. I can understand that an asymptotic analysis may be interesting as it allows one to obtain a first approach to a problem, but if the asymptotic nature of the analysis is fundamental, as it is in this case (the authors mention that positive non-asymptotic results simply are not possible without additional assumptions), then is there hope to finally achieve learning algorithms that can actually be used? I do not see any such hope from this paper, especially in light of the construction of experts that should be fed into the MHA algorithm. Motivation for the objective: This paper lacks sufficient motivation for the formal objective; while the high-level objective is clear, namely, to obtain a strategy that minimizes the average $u$-loss subject to the average $c$-loss being at most $\gamma$, the formal rendering of this objective is not sufficiently motivated. I understand why one would want to use strategies that are $\gamma$-bounded (as defined in Definition 1). However, the authors then further restrict to processes that are $\gamma$-feasible with no motivation for why we should restrict to these processes. How do we know that this restriction is the right one, and not too great of a restriction? If we restrict less, perhaps we would arrive at a different $\mathcal{V}^*$ that is larger than the one arising from $\gamma$-feasible processes. In particular, with additional space, I feel that the authors should be able to properly explain (or at least give some explanation, as none whatsoever is given in the current paper) why we should care about the conditional expectation of $u(y, X_0)$ (conditional on the infinite past). It is paramount to motivate the $\gamma$-feasibility restriction so that $\mathcal{V}^*$ can be understood as a quantity of fundamental interest. Other comments on the paper: While the MHA algorithm seems interesting, without an explanation of what experts are used in the main paper, it is difficult to get a sense of what the algorithm is really doing. The authors should devote some space in the main text explaining the construction of the experts. I wonder if the authors could have provided proof sketches in the main paper for the main results, Theorems 1 and 3, and bought themselves about 3 pages (each of those proofs is about 1.5-2 pages), as well as an exposition that would be clearer to the audience. MINOR COMMENTS I looked through part of the proof of Theorem 1 and it seems fine, but I am not an expert in this area so I am trusting the authors. In the math display between lines 68 and 69, I think the a.s. should not be there. If it should be there, what does it mean? All the randomness has been removed by the outer expectation. Line 131: an histogram'' --> a histogram'' This paper is at times sloppily written. Examples: Line 131, you write $H_{h,k}$ and then later write $k, l = 1, 2, \ldots$, and this swapping of $h$ and $l$ happens throughout the paper. Math display between lines 107 and 108: The minus sign in front of the last summation on the first line should be a plus sign. Optimallity'' -> Optimality'' UPDATE AFTER REBUTTAL I am satisfied with the authors' response regarding the formal framing of the problem and the conditional expectation, provided that they provide details (similar to their rebuttal) in the paper.

Reviewer 2



The paper studies online learning in the multi-objective setting. The paper builds upon previous result by Mahdavi et al, and extends the multi-objective framework to the case of stationary and ergodic processes, allowing for the dependencies among observations. The authors first give a lower bound for all algorithms and present an algorithm which achieves the optimal value. I found the paper to be well-written and relatively easy to follow, despite my lack of familiarity with ergodic theory. The authors succeeded at separating discussion and interpretation of their result from technical details, which made the paper readable. The assumptions on the sequence of outcomes (stationarity and ergodicity) are much weaker from the i.i.d. assumption in the past work. It is hard for me to judge the significance of the paper, as I not familiar with previous work on multi-objective learning, but I found the results interesting, and the theoretical contribution to be strong. In particular, the main results are: - Theorem 1, which states that despite adaptive nature of the prediction strategy, no such strategy can beat the best prediction which knows the (infinite) past. This is something one can expect, but it is nice to see it shown. - Convergence of Algorithm 1, a specific version of exponential weights algorithm, to the optimal solution for stationary and ergodic sequences. I think it would be beneficial to describe the construction of the experts in a bit more detail in the main part of the paper. It is not entirely clear what is the size of the expert set? From what I understand, the set is countable, but not necessarily finite. Can then the algorithm be actually run on this set? Can anything be said about the performance of the algorithm for finite samples (apart from "regret bounds" (9) and (10))? What are the constants C_{k,h}, C'_{k,h} in the proof of Theorem 3? Are they related to the prior probability \alpha assigned to a given "expert"? ------- I have read the rebuttal.

Reviewer 3



The paper addresses prediction of stationary ergodic processes with multiple objective loss functions. In this setting the average loss incurred according to one loss function cannot be exceeded above a certain threshold, while the average loss incurred due to a second loss function needs to be minimized. This problem has been solved for i.i.d. processes with multiple objective loss functions and for stationary ergodic processes with a single objective function. This paper tackles multiple constraints and stationary ergodic processes at the same time. The paper is clear and well written. The results appear to be correct and the problem is well motivated.